EMBO
Molecular Medicine

# The time-of-day of myocardial infarction onset affects healing through oscillations in cardiac neutrophil recruitment

Maximilian J Schloss[1], Michael Horckmans[1], Katrin Nitz[1,2], Johan Duchene[1], Maik Drechsler[1,3,4], Kiril Bidzhekov[1], Christoph Scheiermann[5], Christian Weber[1,3,6], Oliver Soehnlein[1,3,4] & Sabine Steffens[1,3,*]

## Abstract

Myocardial infarction (MI) is the leading cause of death in Western countries. Epidemiological studies show acute MI to be more prevalent in the morning and to be associated with a poorer outcome in terms of mortality and recovery. The mechanisms behind this association are not fully understood. Here, we report that circadian oscillations of neutrophil recruitment to the heart determine infarct size, healing, and cardiac function after MI. Preferential cardiac neutrophil recruitment during the active phase (Zeitgeber time, ZT13) was paralleled by enhanced myeloid progenitor production, increased circulating numbers of CXCR2[hi] neutrophils as well as upregulated cardiac adhesion molecule and chemokine expression. MI at ZT13 resulted in significantly higher cardiac neutrophil infiltration compared to ZT5, which was inhibited by CXCR2 antagonism or neutrophil-specific CXCR2 knockout. Limiting exaggerated neutrophilic inflammation at this time point significantly reduced the infarct size and improved cardiac function.

**Keywords** circadian rhythm; fibrosis; myocardial infarction healing; neutrophils; progenitors

**Subject Categories** Cardiovascular System; Immunology

## Introduction

The incidence of cardiovascular events, such as MI, ischemic stroke, and arrhythmias, exhibits time-of-day dependency in humans, peaking around the sleep-to-wake transition period (Muller *et al*, 1987a,b; Chen & Yang, 2015). The underlying mechanisms for this time-of-day dependency are thought to involve circadian fluctuations of glucocorticoids and catecholamines, blood pressure, heart rate, blood viscosity, and platelet reactivity, thereby predisposing for plaque rupture and thrombus formation (Tofler *et al*, 1987; Chen & Yang, 2015). In addition to the increased prevalence of MI in the morning, experimental and clinical evidence suggests that the outcome after MI exhibits a similar time-of-day dependency. A circadian variation of infarct size has first been described in a mouse model of ischemia/reperfusion, showing significantly larger infarct size, fibrosis, and adverse remodeling after ischemia onset at the sleep-to-wake transition period (Durgan *et al*, 2010). Likewise, several clinical studies reported a correlation between infarct size assessed by peak creatine kinase and the time-of-day of ischemia onset (Suarez-Barrientos *et al*, 2011; Fournier *et al*, 2012; Reiter *et al*, 2012).

In recent years, circadian oscillations of immune cell functions and circulating mediators (e.g. hematopoietic stem cells, glucocorticoids) have emerged (Scheiermann *et al*, 2013). Circulating leukocytes oscillate between blood and peripheral tissue, peaking in mice at ZT5 (where ZT0 refers to lights on and ZT12 to lights off) in the blood and at ZT13 in muscle tissue and bone marrow (Scheiermann *et al*, 2012). In humans, which have an opposing sleep–wake cycle, blood neutrophils oscillate throughout the day with an amplitude of $0.31 \times 10^9$/l and a high point around 8:30 pm (Sennels *et al*, 2011). These fluctuations in immune cell trafficking into tissues coincide with sensitivity to acute inflammatory stimuli, being highest at the beginning of the active phase (Scheiermann *et al*, 2013). Whether these oscillations in immune cell activity occur in the heart after an infarction and which consequences this would have on myocardial healing is unknown.

High numbers of circulating neutrophils are generally considered detrimental for post-MI outcome (Kyne *et al*, 2000; Chia *et al*, 2009). Neutrophils massively infiltrate the ischemic myocardium within the first 24 h post-MI, especially when reopening of the

---

1   Institute for Cardiovascular Prevention (IPEK), Ludwig-Maximilians-University (LMU) Munich, Munich, Germany
2   Max Delbrueck Center for Molecular Medicine in the Helmholtz Association, Berlin, Germany
3   German Centre for Cardiovascular Research (DZHK), partner site Munich Heart Alliance, Munich, Germany
4   Department of Pathology, Amsterdam Medical Center (AMC), Amsterdam, The Netherlands
5   Walter-Brendel-Center of Experimental Medicine, Ludwig-Maximilians-University (LMU) Munich, Munich, Germany
6   Department of Biochemistry, Cardiovascular Research Institute Maastricht (CARIM), Maastricht University, Maastricht, The Netherlands
    *Corresponding author. Tel: +49 89 4400 54674; E-mail: sabine.steffens@med.uni-muenchen.de

occluded coronary artery is achieved (Frangogiannis, 2012). Depleting neutrophils during the reperfusion phase limited acute tissue injury in experimental models (Romson *et al*, 1983; Litt *et al*, 1989). Nevertheless, despite their detrimental role during the acute post-ischemic inflammatory role, a limited number of neutrophils might be important for coordinated resolution of post-MI inflammation and repair (Frangogiannis, 2012). This is supported by our own recent findings that neutrophil depletion in a mouse model of permanent LAD occlusion negatively affects cardiac healing after MI (when induced during the acrophase of neutrophils in the blood, ZT5; Horckmans *et al*, 2016).

In the present study, we raised the question whether the magnitude of neutrophil-driven inflammatory response and quality of the healing response in a murine MI scenario is influenced by circadian oscillations of neutrophil recruitment to the heart. We found that the migration of neutrophils to the heart preferentially occurs during the active phase (ZT13). MI at this time point resulted in significantly higher cardiac neutrophil infiltration, in a CXCR2-dependent manner. Consequently, an ischemic event occurring during the active phase resulted in an exaggerated neutrophilic inflammation and worsened cardiac repair. Limiting neutrophil counts at this time point reduced the infarct size and improved cardiac function. Our findings suggest that the time-of-day of ischemia onset is a critical determinant when considering anti-inflammatory treatments for improving MI outcome.

# Results

### Circadian oscillations of neutrophils in the mouse heart under steady state

We first investigated whether a rhythmic recruitment of neutrophils to the cardiac muscle occurs, as previously reported for skeletal muscle (Scheiermann *et al*, 2012). The analysis of blood counts in resting WT mice confirmed a peak of blood granulocyte counts at ZT5 and lowest levels around ZT13 (Fig 1A). The flow cytometric analysis of digested hearts revealed more than twofold higher neutrophil counts in the myocardium at ZT13 compared to ZT5 (Fig 1B). We also performed immunohistological staining which revealed the presence of a limited number of neutrophils in the healthy myocardium, with markedly more cells at ZT13 compared to ZT5 (Fig 1C). In support of enhanced adhesion and subsequent extravasation of neutrophils into the cardiac tissue, cardiac ICAM-1 and VCAM-1 adhesion molecule mRNA expression increased during the active phase (ZT13-17; Fig 1D), which was paralleled by enhanced mRNA levels of chemokines mediating neutrophil chemotaxis, that is, CXCL1, CXCL2, CXCL5, CCL3, and CCL5 (Fig 1D). Analysis of the corresponding chemokine receptor for CXCL1, CXCL2, and CXCL5 on circulating and cardiac neutrophils, CXCR2, revealed an increased expression level in both compartments at ZT13 (Fig 1E).

### Increased cardiac neutrophil infiltration after MI during active phase is associated with enhanced myelopoiesis and neutrophil mobilization

We next investigated whether onset of MI during the peak of cardiac neutrophil counts would lead to increased neutrophil recruitment.

Indeed, mice subjected to LAD occlusion at ZT13 had higher neutrophil counts in the heart 12–24 h post-MI compared to ZT5 infarcts (Fig 2A). Of note, 12 h post-MI induced at ZT13 is the time point when oscillating neutrophil counts in tissues are lowest under steady-state condition. Neutrophils were mobilized from the bone marrow, resulting in a significant decrease in neutrophil counts in femurs 12–24 h post-MI, which was much more dramatic in ZT13-infarcted mice (Fig 2B). In line with enhanced cardiac infiltration after ZT13 MI, blood counts of granulocytes tended to be lower in mice with ZT13 MI compared to ZT5 MI (12–24 h post-MI), albeit the differences were not significant (Fig 2C). To confirm the relevance of circadian time and light for the reported effects on cardiac neutrophil recruitment, we entrained mice to an inverted light cycle. Under these conditions, we found a similar increase in cardiac neutrophil recruitment and enhanced bone marrow mobilization 24 h after ZT13 MI compared to ZT5 MI (Fig 2D).

To better understand the mechanisms of neutrophil mobilization, we further assessed progenitor numbers in the bone marrow. We found higher baseline granulocyte–monocyte progenitor (GMP) counts in the bone marrow at ZT13 compared to ZT5, which were even higher 24 h after MI (Fig 2E). In agreement with published data (Mendez-Ferrer *et al*, 2008), bone marrow levels of the retention signal CXCL12 decreased from ZT5 to ZT13; however, there was no further reduction in the bone marrow of ZT13-operated mice (Fig 2F). We therefore reasoned that enhanced neutrophil mobilization after MI at ZT13 might be triggered by enhanced circulating levels of neutrophil chemoattractants. We found a remarkable upregulation of TNF-α, CXCL1, CXCL2, CCL3, CCL5, and G-CSF levels in the plasma of ZT13- versus ZT5-operated mice 24 h post-MI (Fig 2G). Of note, no difference in chemokine and cytokine plasma levels between ZT5 and ZT13 was found without infarction, and there was no induction of CXCL5 plasma levels after MI.

### MI during active phase leads to larger infarcts and reduced cardiac function

In support of the concept that exaggerated neutrophil presence in ZT13-infarcted hearts results in increased cardiac damage, mice subjected to LAD occlusion at ZT13 resulted in larger infarcts compared to mice with ZT5 MI (Fig 3A). The morphometric analysis was confirmed by elevated plasma levels of troponin I (Fig 3B) and higher numbers of dead cardiomyocytes after ZT13 MI (Fig 3C). A similar increase in troponin I levels was observed in infarcted mice with shifted light cycle (Appendix Fig S1). The mortality rate after ZT13 MI was significantly higher than after ZT5 MI, with increased incidence of ventricular rupture in this group (Fig 3D).

In agreement with larger infarcts, ZT13 infarcts had more myofibroblasts 7 days after MI (Fig 3E and Appendix Fig S2A), resulting in an increased area of fibrosis (Fig 3F). However, the anterior wall thickness was significantly reduced in ZT13 infarcts. More detailed analysis of the collagen composition revealed lower density of thicker collagen type I fibers in ZT13 infarcts (Fig 3G and Appendix Fig S2B). Thus, insufficient stabilization of the infarct scar might contribute to the increased incidence of ventricular rupture after ZT13 MI. Mice with ZT13 infarcts also had significantly lower ejection fractions at day 3 to day 14 compared to mice subjected to MI at ZT5, as well as significant increase in the left ventricular end-diastolic and end-systolic volume (Fig 3H).

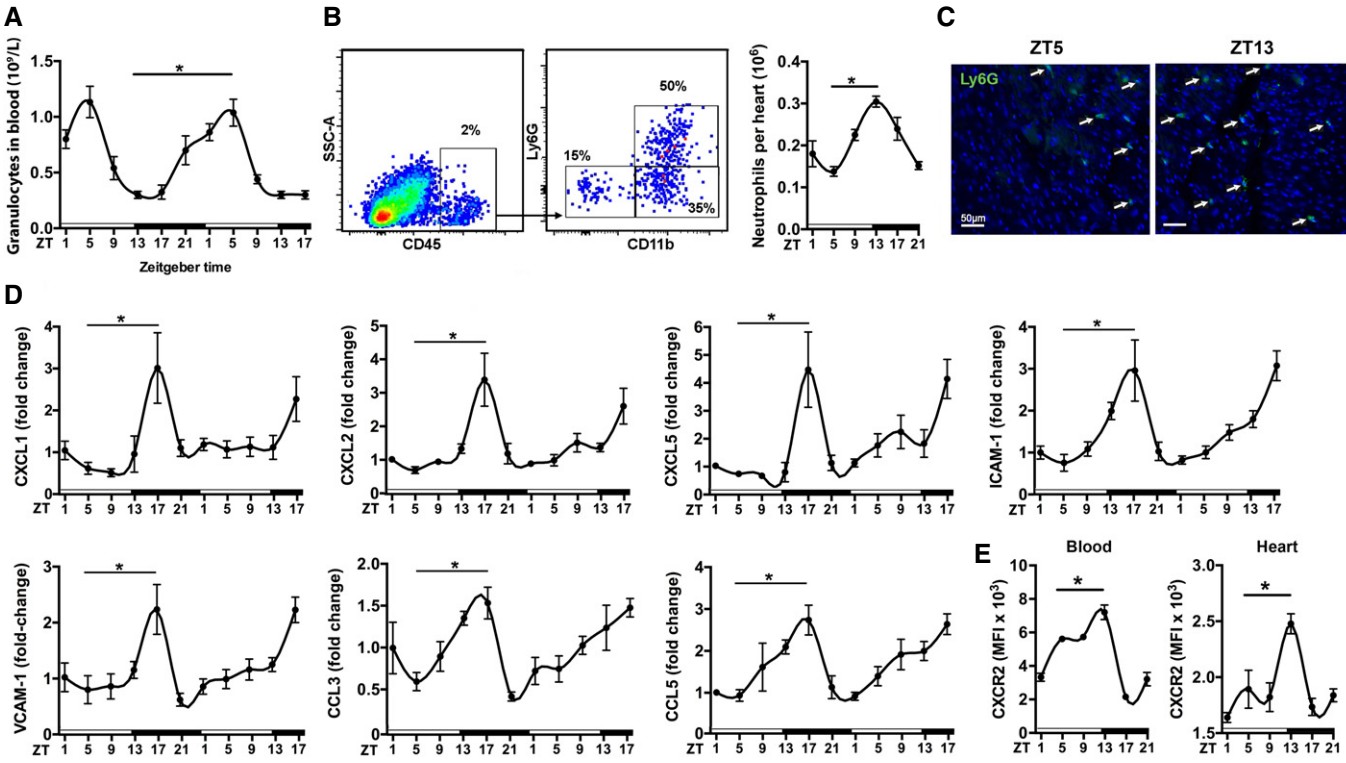

**Figure 1. Circadian oscillations of neutrophils, adhesion molecules, and chemokines in the mouse heart at steady state.**

A  Baseline blood counts of granulocytes. One-way ANOVA; $n$ = 8 mice per ZT; *$P$ = 0.0001 ZT5 versus ZT13.

B  Flow cytometric quantification of neutrophils in digested hearts. The representative dot plots show the gating strategy for cardiac neutrophils (CD45$^+$Ly6G$^+$CD11b$^+$) at ZT5. One-way ANOVA; $n$ = 5 mice per ZT; *$P$ = 0.0001 ZT5 versus ZT13.

C  Representative immunostainings for neutrophils in the myocardium (left ventricle), identified as Ly6G positive (20× magnification).

D  Cardiac mRNA expression levels normalized to HPRT. One-way ANOVA; $n$ = 5 mice per ZT; ZT5 versus ZT17: *$P$ = 0.0064 (CXCL1), *$P$ = 0.0007 (CXCL2), *$P$ = 0.0007 (CXCL5), *$P$ = 0.0001 (ICAM-1), *$P$ = 0.0009 (VCAM-1), *$P$ = 0.0181 (CCL3), *$P$ = 0.0360 (CCL5).

E  Mean fluorescence intensity (MFI) of CXCR2 expression by neutrophils in blood and heart. One-way ANOVA; $n$ = 3 mice for ZT1, ZT17, ZT21 and $n$ = 5 for ZT5, ZT9, ZT13; ZT5 versus ZT13: *$P$ = 0.0425 (blood), *$P$ = 0.0078 (heart).

Data information: All data are expressed as mean ± SEM.

### Reduction in neutrophil-mediated inflammation during active phase preserves cardiac function after MI

To confirm that the larger infarct size and decreased cardiac function after ZT13 MI are mainly driven by an exaggerated neutrophilic inflammatory response, we investigated whether systemic reduction in neutrophils would limit this process. Mice subjected to MI at ZT13 received neutrophil-depleting Ly6G antibody (Fig 4A), resulting in a significant reduction in cardiac neutrophils (Fig 4B), approximately to the levels observed at ZT5 reported in Fig 3, and lower plasma levels of pro-inflammatory cytokine TNF-α 24 h after MI (Fig 4C). Mice with ZT13 MI receiving anti-Ly6G had significantly smaller infarcts and lower plasma troponin I levels compared to isotype-treated controls, whereas anti-Ly6G treatment of mice with ZT5 MI had no effect (Fig 4D). This is in agreement with our recently published data (Horckmans *et al*, 2016). Moreover, ZT13 infarcts of Ly6G-treated mice had less fibrosis and thicker left ventricular anterior wall (Fig 4E) with significantly more collagen type I fibers (Fig 4F), and a higher ejection fraction as well as less ventricular dilatation (Fig 4G).

### Enhanced cardiac neutrophil recruitment during the active phase is CXCR2 dependent

To further explore the underlying mechanisms of circadian neutrophil recruitment to the myocardium, we focused on CXCR2, as its expression levels coincided with the peak of neutrophils in the heart (Fig 1E). There is emerging evidence that circulating neutrophils are heterogeneous, due to aging and replenishment from the bone marrow (Casanova-Acebes *et al*, 2013). Previous *in vitro* findings have suggested that aged neutrophils exhibit reduced chemotactic activity and ability to respond to inflammatory stimuli (Whyte *et al*, 1993). Similar to the oscillations at baseline, we found significantly higher percentage of CXCR2$^{hi}$-expressing neutrophils in the blood of mice subjected to MI at ZT13 compared to ZT5 MI, with highest levels 8 h after ZT13 MI (Fig 5A). All cardiac neutrophils were CXCR2 positive regardless of the time point of LAD occlusion (measured 12 h post-MI at ZT5 versus ZT13); however, the level of CXCR2 surface expression was much higher at ZT13 compared to ZT5 (Fig 5B). In order to clarify the causal relationship between time-of-day-dependent changes in CXCR2 expression levels on circulating neutrophils and their recruitment to the ischemic

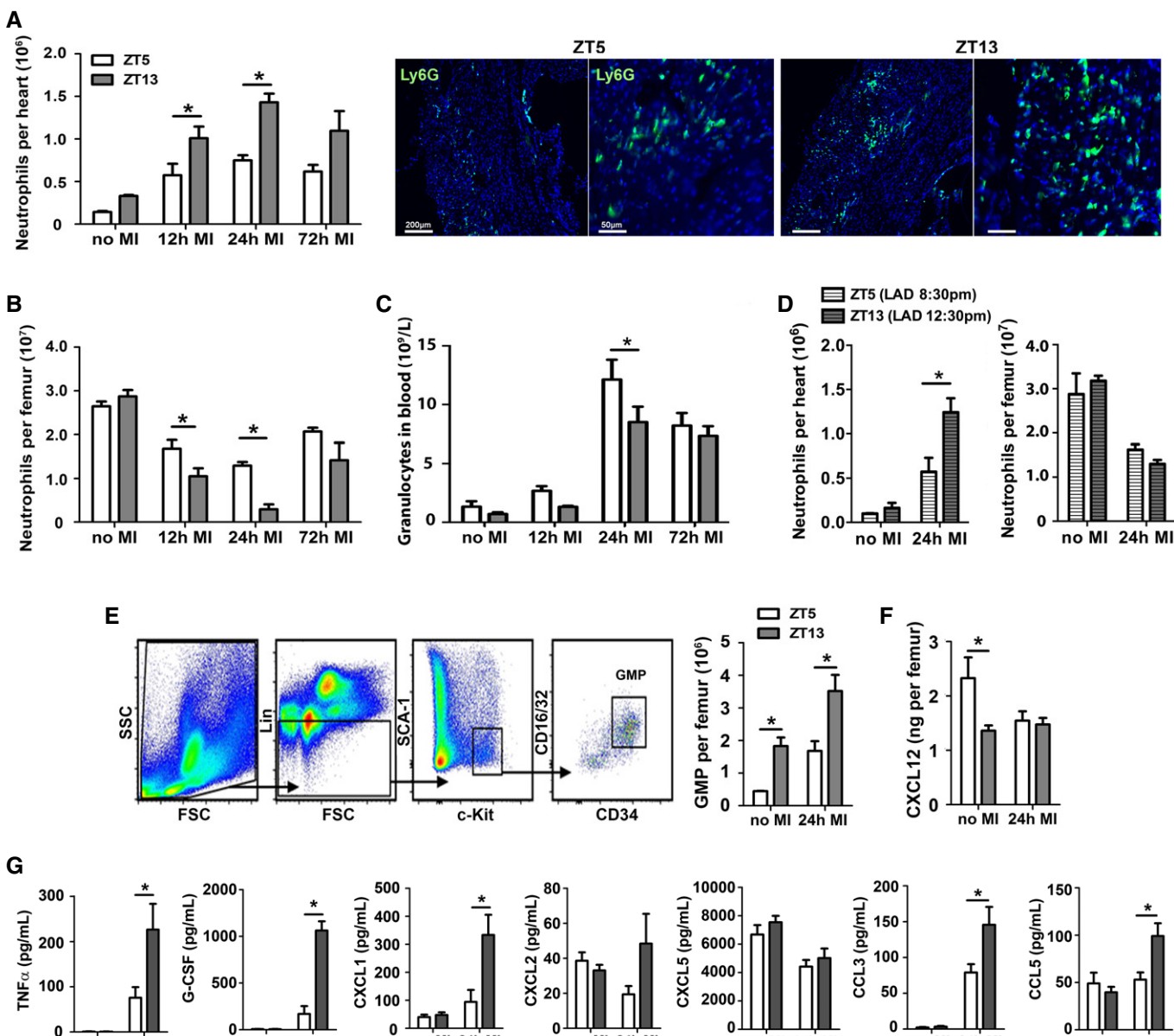

**Figure 2.   Inflammatory response after MI during active (ZT13) or resting phase (ZT5).**

A   Flow cytometric analysis of neutrophils in hearts and representative immunostainings for neutrophils in the infarct area, identified as Ly6G positive (5× and 20× magnifications). Two-way ANOVA followed by Bonferroni *post hoc* test; $n = 5$ mice for no MI, $n = 5$ for 12 h post-MI, $n = 3$ for 24 h post-MI, and $n = 3$ for 72 h post-MI in both ZT groups; ZT5 versus ZT13: *$P = 0.0161$ (12 h MI), *$P = 0.003$ (24 h MI).

B   Flow cytometric analysis of neutrophils in bone marrow. Two-way ANOVA followed by Bonferroni *post hoc* test; $n = 5$ mice for no MI, $n = 5$ for 12 h post-MI, $n = 4$ for 24 h post-MI, and $n = 4$ for 72 h post-MI in both ZT groups; ZT5 versus ZT13: *$P = 0.0471$ (12 h MI), *$P = 0.0035$ (24 h MI).

C   Blood counts of granulocytes. Two-way ANOVA followed by Bonferroni *post hoc* test; $n = 9$ mice for no MI, $n = 10$ for 12 h post-MI, $n = 13$ for 24 h post-MI, and $n = 10$ for 72 h post-MI in both ZT groups; ZT5 versus ZT13: *$P = 0.0364$ (24 h MI).

D   Flow cytometric analysis of neutrophils in hearts and bone marrow under inverted light cycle conditions. Two-way ANOVA followed by Bonferroni *post hoc* test; $n = 3$ mice for no MI in both ZT groups, $n = 5$ for ZT5 and $n = 6$ for ZT13 at 24 h post-MI; ZT5 versus ZT13: *$P = 0.0147$ (24 h MI).

E   Representative gating strategy for GMPs in the bone marrow, identified as lineage negative (CD11b⁻, Gr1⁻, B220⁻, CD3⁻, and Ter119⁻) and Sca-1⁻, c-kit⁺, CD16/32⁺, and CD34⁺. Flow cytometric quantification of GMP in the bone marrow. Two-way ANOVA followed by Bonferroni *post hoc* test; $n = 4$ mice for no MI and 24 h post-MI in both ZT groups; ZT5 versus ZT13: *$P = 0.0077$ (no MI), *$P = 0.0013$ (24 h MI).

F   CXCL12 levels in bone marrow lavage. Two-way ANOVA followed by Bonferroni *post hoc* test; $n = 7$ mice for no MI and $n = 6$ for 24 h post-MI in both groups; ZT5 versus ZT13: *$P = 0.0077$ (no MI), *$P = 0.0013$ (24 h MI).

G   Plasma levels of pro-inflammatory cytokines and chemokines. Two-way ANOVA followed by Bonferroni *post hoc* test; $n = 11$ mice for no MI in both ZT groups, $n = 7$ for ZT5 and $n = 8$ for ZT13 at 24 h post-MI; ZT5 versus ZT13: *$P = 0.0271$ (CXCL12, no MI), *$P = 0.0108$ (TNF-α, 24 h MI), *$P = 0.001$ (G-CSF, 24 h MI), *$P = 0.005$ (CXCL1, 24 h MI), *$P = 0.0005$ (CXCL2, 24 h MI), *$P = 0.0016$ (CCL3 24 h MI), *$P = 0.0144$ (CCL5, 24 h MI).

Data information: All data are expressed as mean ± SEM.

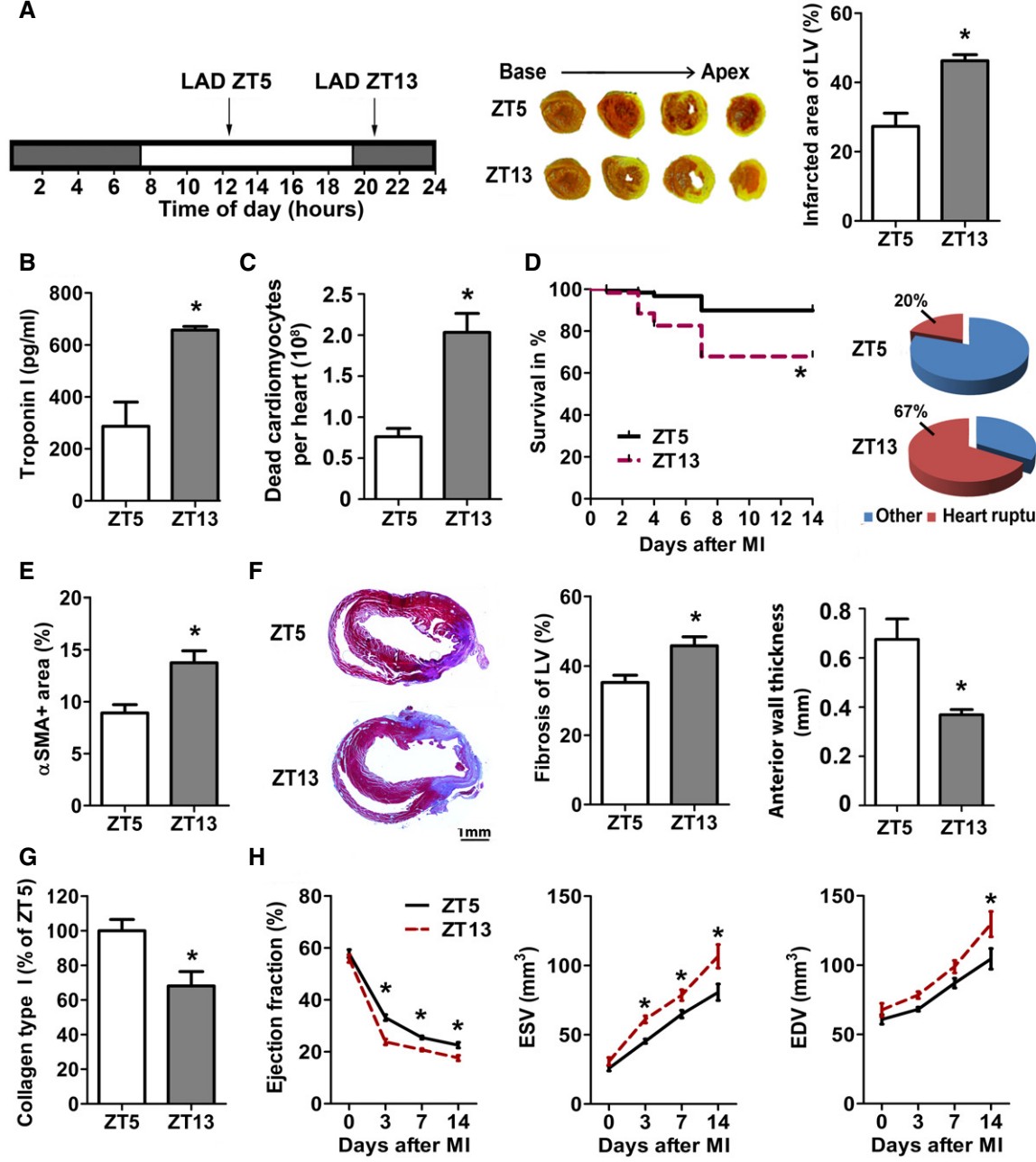

**Figure 3. MI during active phase (ZT13) leads to infarct expansion and reduced cardiac function.**

A   Permanent LAD occlusion was performed at ZT5 or ZT13. TTC staining (white, infarct; red, vital myocardium) and quantification of infarct size normalized to the left ventricle (LV). Student's *t*-test; *n* = 4 mice for ZT5 and *n* = 5 for ZT13 MI; *P = 0.0041.

B   Plasma troponin I levels 24 h after MI. Student's *t*-test; *n* = 4 mice for ZT5 and *n* = 5 for ZT13 MI; *P = 0.0007.

C   Flow cytometric analysis of dead cardiomyocytes (CD45⁻, Zombie⁺) 24 h after MI. Student's *t*-test; *n* = 3 mice in both groups; *P = 0.0072.

D   Survival rates after MI and cause of death. Log-rank test; *n* = 87 mice in both groups; *P = 0.0006.

E   Myofibroblasts within infarcts were quantified by alpha-smooth muscle actin (αSMA) staining as ratio between stained and total area of random fields. Student's *t*-test; *n* = 4 mice in both groups; *P = 0.0134.

F   Masson's trichome staining of fibrosis (blue, collagen; red, vital myocardium) and quantification relative to total LV (*P = 0.0095) as well as LV anterior wall thickness (*P = 0.0068) 7 days after MI. Student's *t*-test; *n* = 6 mice for ZT5 and *n* = 7 for ZT13 MI.

G   Analysis of relative collagen type I content identified by Sirius Red staining 7 days after MI. Student's *t*-test; *n* = 4 mice for ZT5 and *n* = 5 for ZT13 MI; *P = 0.0175.

H   Echocardiographic assessment of ejection fraction (EF), end-systolic volume (ESV), and end-diastolic volume (EDV). Two-way ANOVA; *n* = 6 mice for no MI for both groups, *n* = 6 for ZT5 and *n* = 9 for ZT13 at 72 h post-MI, *n* = 8 for both groups at 7 days post-MI, and *n* = 7 for ZT5 and *n* = 5 for ZT13 at 14 days post-MI; ZT5 versus ZT13: *P = 0.0001 (EF, 3 days), *P = 0.0042 (EF, 7 days), *P = 0.0121 (EF, 14 days); *P = 0.0253 (ESV, 3 days), *P = 0.0421 (ESV, 7 days), *P = 0.0005 (ESV, 14 days); *P = 0.0053 (EDV, 14 days).

Data information: All data are expressed as mean ± SEM.

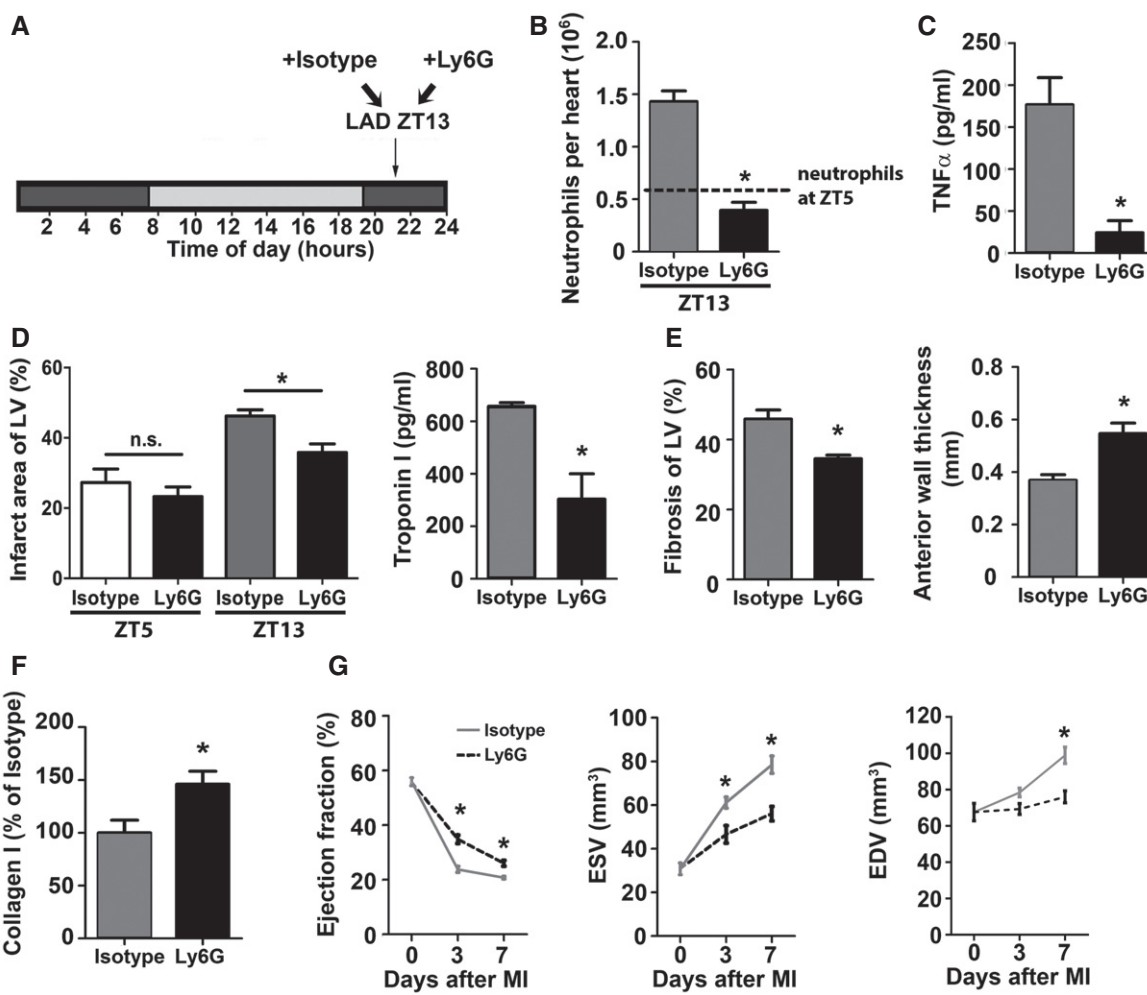

**Figure 4. Limiting neutrophilic inflammation during active phase (ZT13) reduces MI damage.**

A   Permanent LAD occlusion was performed at ZT13 followed by injection of isotype or Ly6G antibody 45 min after surgery and then every 24 h.

B   Flow cytometric analysis of cardiac neutrophils 24 h after ZT13 MI. The dotted line indicates cardiac neutrophil counts 24 h after ZT5 MI, as shown in Fig 2A. Student's t-test; $n = 3$ mice for isotype and $n = 5$ for Ly6G injected mice; *$P = 0.0004$.

C   Plasma TNF-α levels 24 h after ZT13 MI. Student's t-test. $n = 8$ independent samples for isotype and $n = 6$ for Ly6G injected mice; *$P = 0.0019$.

D   Infarct size relative to left ventricular area (LV) and plasma troponin I levels 24 h after ZT5 or ZT13 MI. Student's t-test; for infarct size, $n = 3$ mice in both groups for ZT5 and $n = 4$ mice for isotype and $n = 3$ mice for Ly6G at ZT13; isotype versus Ly6G: *$P = 0.0158$ (ZT13). For troponin levels, $n = 8$ mice for isotype and $n = 6$ mice for Ly6G with ZT13 MI; *$P = 0.0012$.

E   Masson's trichrome staining of fibrosis (blue, collagen; red, vital myocardium) and quantification relative to total area of the LV (*$P = 0.0084$); morphometric quantification of the LV anterior wall thickness (*$P = 0.0027$) 7 days after ZT13 MI. Student's t-test; $n = 5$ mice for isotype and $n = 4$ mice for Ly6G.

F   Analysis of collagen type I fibers within infarcts identified by Sirius Red staining 7 days after ZT13 MI. Student's t-test; $n = 5$ mice for isotype and $n = 4$ mice for Ly6G; *$P = 0.0375$.

G   Echocardiographic measurement of ejection fraction (EF), end-systolic volume (ESV) and end-diastolic volume (EDV) before and after ZT13 MI. Two-way ANOVA; $n = 6$ mice for no MI in both groups, $n = 9$ for isotype and $n = 5$ for Ly6G at 72 h post-MI, and $n = 8$ for isotype and $n = 4$ for Ly6G at 7 days post-MI. Isotype versus Ly6G: *$P = 0.0001$ (EF, 3 days), *$P = 0.0227$ (EF, 7 days); MI *$P = 0.0133$ (ESV, 3 days), *$P = 0.0004$ (ESV, 7 days); *$P = 0.0017$ (EDV, 7 days).

Data information: All data are expressed as mean ± SEM.

myocardium, we did functional blocking experiments with CXCR2 antagonist SB225002. CXCR2 antagonism induced a massive reduction in CXCR2$^{hi}$-expressing blood neutrophils in mice with ZT13 MI (Fig 5C). Remarkably, the percentages of CXCR2$^{hi}$ blood neutrophils in the ZT5 MI group remained unchanged after SB225002 treatment (Fig 5C). Likewise, the CXCR2 antagonist did not affect neutrophil numbers in bone marrow and heart of mice with ZT5 MI (Fig 5D and E), whereas cardiac neutrophil recruitment

in ZT13 infarcts was significantly inhibited by CXCR2 antagonism (Fig 5D). This was paralleled by reduced mobilization from the bone marrow in these mice (Fig 5E).

To validate the role of CXCR2 in circadian rhythm-dependent cardiac neutrophil recruitment, we performed additional experiments with lethally irradiated wild-type mice transplanted with Mrp8-Cre-CXCR2flox bone marrow. Transplantation with bone marrow from CXCR2flox littermates served as wild-type control. We

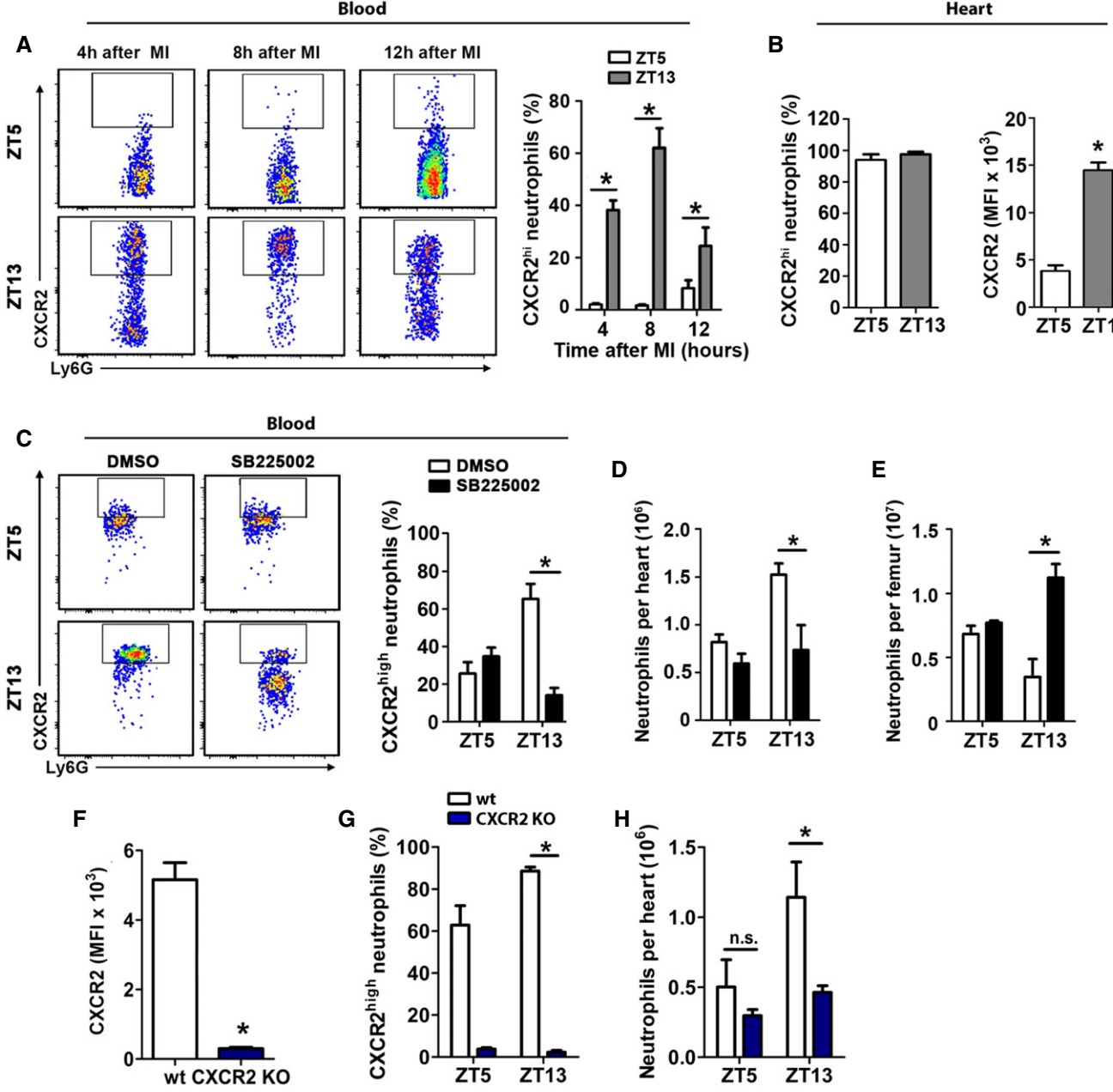

**Figure 5. Antagonism or deficiency of CXCR2 inhibits enhanced cardiac neutrophil accumulation during active phase (ZT13).**

A Representative flow cytometric analysis and quantification of CXCR2[high] neutrophils in the blood after ZT5 and ZT13 MI. Two-way ANOVA followed by Bonferroni *post hoc* test; *n* = 4 mice for all time points in both groups; ZT5 versus ZT13: \**P* = 0.0001 (4 h), \**P* = 0.0001 (8 h), \**P* = 0.0447 (12 h).

B Percentage and mean fluorescence intensity (MFI) of CXCX2 expression by cardiac neutrophils 12 h post-MI after ZT5 and ZT13 MI. Student's *t*-test; *n* = 4 mice in both groups; \**P* = 0.0001.

C Percentage of CXCR2[high] neutrophils in the blood 24 h after ZT5 or ZT13 MI in mice receiving CXCR2 antagonist SB225002 or vehicle. Two-way ANOVA followed by Bonferroni *post hoc* test; *n* = 4 mice in both groups; DMSO versus SB225002: \**P* = 0.0005 (ZT13).

D Flow cytometric quantification of neutrophils in hearts 24 h after ZT5 or ZT13 MI in mice receiving CXCR2 antagonist SB225002 or vehicle. Two-way ANOVA followed by Bonferroni *post hoc* test; *n* = 4 mice in both groups; DMSO versus SB225002: \**P* = 0.0079 (ZT13).

E Flow cytometric quantification of neutrophils in bone marrow 24 h after ZT5 or ZT13 MI in mice receiving CXCR2 antagonist SB225002 or vehicle. Two-way ANOVA followed by Bonferroni *post hoc* test; *n* = 4 mice in both groups; DMSO versus SB225002: \**P* = 0.0002 (ZT13).

F MFI of CXCR2 expression at ZT5 in the blood 24 h after MI in wild-type (WT) and CXCR2 KO mice. Student's *t*-test. *n* = 6 mice in both groups; \**P* = 0.0001.

G Percentage of CXCR2[hi] blood neutrophils 24 h after ZT5 or ZT13 MI in WT and CXCR2 KO mice. Two-way ANOVA followed by Bonferroni *post hoc* test; for ZT5 *n* = 6 mice in both groups, for ZT13 *n* = 7 WT mice and *n* = 5 CXCR2 KO mice; WT versus CXCR2 KO: \**P* = 0.0001 (ZT13).

H Flow cytometric quantification of neutrophils in hearts 24 h after MI in ZT5 and ZT13-operated WT and CXCR2 KO mice. Two-way ANOVA followed by Bonferroni *post hoc* test; for ZT5 *n* = 6 mice in both groups, for ZT13 *n* = 7 WT mice and *n* = 5 CXCR2 KO mice; WT versus CXCR2 KO: \**P* = 0.0461 (ZT13), ns = not significant.

Data information: All data are expressed as mean ± SEM.

first validated the knockdown of CXCR2 on neutrophils (Fig 5F) and confirmed that increased numbers of CXCR2hi neutrophils at ZT13 were also found in control mice transplanted with wild-type CXCR2 bone marrow, assessed 24 h after MI (Fig 5G). In agreement with the findings of pharmacological CXCR2 antagonism, neutrophil CXCR2 deficiency blunted the time-of-day-dependent differences in cardiac neutrophil counts after ZT5 and ZT13 MI (Fig 5H).

### CXCR2 antagonism administered shortly before reperfusion inhibits exaggerated cardiac neutrophil recruitment at ZT13

Finally, we aimed to validate the effect of CXCR2 blockade in a clinically more relevant scenario and therefore subjected mice to transient LAD occlusion (Fig 6A). Similar to the effects observed in the permanent occlusion model, we found a higher percentage of CXCR2hi blood neutrophils at ZT13 compared to ZT5, assessed 24 h after reperfusion (Fig 6B). Moreover, vehicle-treated mice 24 h after ZT13 MI had significantly higher neutrophil counts in the myocardium than ZT5 infarcted mice, as well as enhanced mobilization from the bone marrow (Fig 6C and D). Administration of the CXCR2 antagonist after LAD occlusion 5 min before reopening the vessel prevented excessive neutrophil recruitment at ZT13 (Fig 6C and D), suggesting a potential clinical benefit for targeting this receptor.

## Discussion

In this study, we provide evidence that the time-of-day determines the severity of MI damage and outcome. This is due to oscillations of cardiac neutrophil recruitment regulated by modulation of their CXCR2 receptor expression levels. Our findings thereby reveal a

potential explanation for the poorer outcomes in subjects with MI in the early morning hours.

Circadian oscillations of leukocytes between blood and peripheral tissue have been previously reported, peaking at ZT5 in the blood of mice and at ZT13 in skeletal muscle (Scheiermann et al, 2013). As humans have an opposing sleep–wake cycle, the peak of blood neutrophils is around 8:30 pm (Sennels et al, 2011). Here, we extended these findings to the murine myocardium: At baseline, we found twofold higher numbers of neutrophils in the heart at ZT13 compared with ZT5. This is facilitated by enhanced cardiac expression of adhesion molecules and neutrophil chemoattractants, that is, CXCL1, CXCL2, CXCL5, CCL3, and CCL5 at this time point. Interestingly, the circadian modulation of chemokine expression was only detectable in the heart, but not systemically, suggesting a local clock regulating chemokine expression in the myocardium. A similar mechanism has been recently highlighted by Gibbs et al who identified a local pulmonary epithelial cell clock controlling neutrophil recruitment to the lung under inflammatory conditions (Gibbs et al, 2014). Consequently, if an infarct occurs at ZT13, more leukocytes are present in the heart to respond locally to the ischemic injury by releasing pro-inflammatory mediators in order to attract more inflammatory cells into the infarcted area. This might contribute to the enhanced inflammatory response observed in ZT13 infarcts.

Blood neutrophils follow rhythmic cycles of release and migration back to the bone marrow, maintained by circadian changes in bone marrow stromal CXCL12 production and upregulation of CXCR4 by aged neutrophils for their clearance (Casanova-Acebes et al, 2013). These aged CXCR4hi neutrophils concomitantly decrease L-selectin (CD62L) (Casanova-Acebes et al, 2013). Aged neutrophils are thought to exhibit different migratory and pro-inflammatory properties (Whyte et al, 1993); however, more recent data reported enhanced pro-inflammatory activity properties of in vivo aged

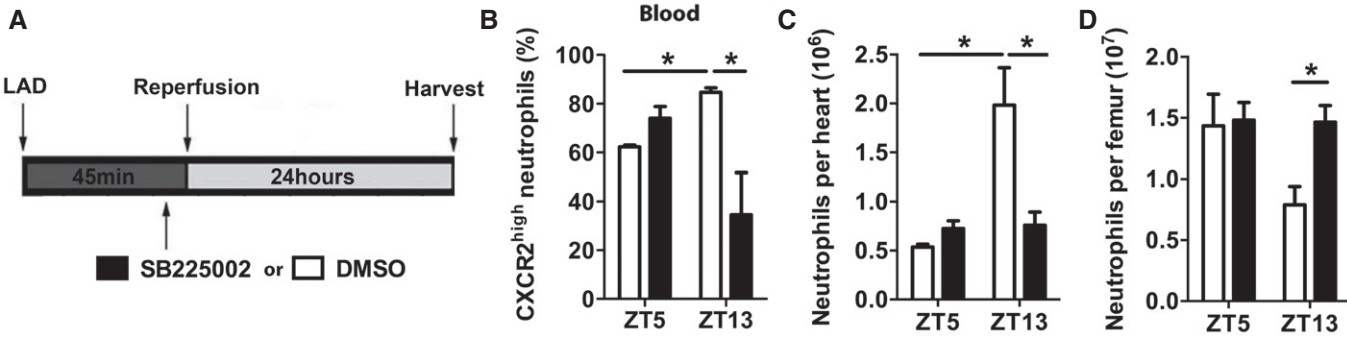

**Figure 6.  Antagonism of CXCR2 inhibits enhanced cardiac neutrophil accumulation after ischemia/reperfusion during active phase (ZT13).**

A   Schematic representation of transient ischemia and reperfusion protocol, performed at ZT5 and ZT13. The CXCR2 antagonist SB225002 or vehicle was injected 5 min before reopening the LAD.

B   Percentage of CXCR2hi neutrophils in the blood 24 h after ZT5 or ZT13 MI in mice receiving CXCR2 antagonist SB225002 or vehicle. Two-way ANOVA followed by Bonferroni post hoc test; n = 5 mice in both groups at ZT5, n = 5 mice for vehicle, and n = 6 mice for SB225002 at ZT13; DMSO versus SB225002: *P = 0.0046 (ZT13); ZT5 versus ZT13: *P = 0.0153 (DMSO).

C   Flow cytometric quantification of cardiac neutrophils 24 h after ZT5 or ZT13 MI in mice receiving CXCR2 antagonist SB225002 or vehicle. Two-way ANOVA; n = 5 mice in both groups at ZT5, n = 5 mice for vehicle, and n = 6 mice for SB225002 at ZT13; DMSO versus SB225002: *P = 0.0006 (ZT13); ZT5 versus ZT13: *P = 0.0001 (DMSO).

D   Flow cytometric quantification of neutrophils in bone marrow 24 h after ZT5 or ZT13 MI in mice receiving CXCR2 antagonist SB225002 or vehicle. Two-way ANOVA followed by Bonferroni post hoc test; n = 5 mice in both groups at ZT5, n = 5 mice for vehicle, and n = 6 mice for SB225002 at ZT13; DMSO versus SB225002: *P = 0,0486 (ZT13).

Data information: All data are expressed as mean ± SEM.

    

neutrophils (Zhang *et al*, 2015). In support of reduced migratory capacity of aged neutrophils, we found a low percentage of circulating CXCR2[hi] neutrophils at ZT5 at steady state, when high numbers of aged CXCR4[hi] CD62L[lo] neutrophils are present in the circulation (Casanova-Acebes *et al*, 2013). Conversely, circulating neutrophils at ZT13 MI, the time point with lowest numbers of aged neutrophils in the blood, were mostly CXCR2[hi] positive. Similar patterns were observed after MI. Our blocking and genetic knockout experiments demonstrated the requirement of CXCR2 for enhanced cardiac neutrophil recruitment at the beginning of the active phase. Of note, CXCR2 antagonism or neutrophil-specific knockout did not generally block neutrophil recruitment at both time points, but only prevented the accelerated infiltration into the myocardium at ZT13 compared to ZT5. Accordingly, it was previously described that bone marrow CXCR2 deficiency did not affect neutrophil infiltration after ischemia/reperfusion (Liehn *et al*, 2013). Thus, additional neutrophil chemoattractants and receptors seem to be involved in cardiac neutrophil recruitment in a circadian-independent manner, whereas the CXCR2 ligand/receptor axis follows rhythmic cycles of expression levels in the myocardium or circulating neutrophils, respectively.

Under homeostatic conditions, we found that GMP numbers increased in the bone marrow at ZT13, which is in agreement with previous findings revealing a rhythmic modulation of the hematopoietic niche (Smaaland *et al*, 1992; Mendez-Ferrer *et al*, 2008). Aged neutrophils return from the circulation back into the bone marrow and are eliminated, thereby providing a signal for regulating the homeostatic release of their own precursors (Casanova-Acebes *et al*, 2013).

The retention and release of neutrophils is tightly regulated by chemokines and their cognate receptors (Kolaczkowska & Kubes, 2013). Bone marrow stromal cells produce CXCL12, which provides a retention signal for hematopoietic cells expressing high levels of CXCR4. Circadian reductions in CXCL12 in the bone marrow correlate with oscillations of hematopoietic progenitor cells in the circulation (Mendez-Ferrer *et al*, 2008). G-CSF, which is upregulated after MI, is known to decrease CXCL12 in the bone marrow in order to facilitate neutrophil mobilization. Surprisingly, there was no further decrease in CXCL12 levels after ZT13 MI despite massive upregulation of G-CSF. A possible explanation is that the CXCL12 levels reach lowest levels at ZT13; thus, no additional decrease in a post-MI inflammatory situation may occur. Instead, the massive release of neutrophils from the bone marrow 24h after ZT13 MI might be explained by the strong increase in chemokine levels in the plasma.

A well-balanced inflammatory response is needed, as dying cardiomyocytes must be removed by phagocytes and replaced by fibrous tissue, since mammals cannot regenerate cardiac tissue (Frangogiannis, 2012). In addition to monocytes/macrophages (Nahrendorf *et al*, 2010), a sufficient number of neutrophils is certainly needed for favorable MI healing (Frangogiannis, 2012). In support of this hypothesis, we have recently found that neutrophils improve cardiac healing after MI by influencing macrophage polarization toward a "reparative" phenotype (Horckmans *et al*, 2016). However, an exaggerated neutrophilic inflammation, as observed after ZT13 MI, generates an environment in which a physiological and beneficial wound healing is impaired. The consequence is an insufficient stabilization of the infarct area by collagen fibers, elevated risk for ventricular rupture and worsening of cardiac function.

Finally, we may speculate that oscillations of pro-inflammatory Ly6C[hi] monocytes (Nguyen *et al*, 2013) contribute to the enhanced inflammatory response after ZT13 MI. Indeed, we found elevated numbers of monocytes in the blood and heart at ZT13 compared to ZT5 under homeostatic conditions; however, monocyte numbers significantly increased only 3 days after MI (data not shown). Neutrophils represent the predominant innate immune cell population that massively infiltrate the infarcted myocardium within the first hours, and we found that limiting neutrophil influx with depleting antibody was successful to prevent excessive cardiac damage at ZT13.

Our observations in this mouse model have certainly limitations as permanent coronary ligation does not reflect the predominant situation in acute MI patients, in which catheter treatment is the gold standard. Therefore, we repeated a key experiment in the ischemia/reperfusion model, thereby confirming a potential clinical relevance for targeting CXCR2 in situations of exaggerated neutrophil infiltration. Our data may provide an explanation for the worse outcomes found in patients suffering an acute MI in the early morning hours.

In conclusion, our findings suggest that the time-of-day of ischemia onset is a critical determinant when considering anti-inflammatory treatments targeting neutrophils for improving MI outcome.

## Materials and Methods

### Animal model of myocardial infarction

Adult (8–10 week old) female C57BL/6J wild-type mice (Janvier Labs, France) were housed for at least 2 weeks under controlled conditions in a 12-h light/12-h dark cycle with lights on at 7:30 am (ZT0) and lights off at 7:30 pm (ZT12). Littermates were randomized and subjected to permanent ligation of the left anterior descending coronary artery (LAD) at ZT5 (12:30 pm) or ZT13 (8:30 pm). Mice were anesthetized with midazolam (5 mg/kg), medetomidine (0.5 mg/kg), and fentanyl (0.05 mg/kg), intubated, and ventilated with a MiniVent mouse ventilator (Harvard Apparatus). A left thoracotomy was performed in the 4[th] left intercostal space, and the pericardium was incised. MI was induced by permanent ligation of the LAD proximal to its bifurcation from the main stem with monofilament nylon 8-0 sutures (Ethicon, Somerville, USA). The chest wall and skin were closed with 5-0 nylon sutures (Ethicon). After surgery, naloxone (1.2 mg/kg), flumazenil (0.5 mg/kg), and atipamezolhydrochlorid (2.5 mg/kg) were injected to reverse the effect of anesthesia. Postoperative analgesia (buprenorphine, 0.1 mg/kg) was given subcutaneously for the first 12 h after surgery. Sham-operated animals were submitted to the same surgical protocol as described but without LAD occlusion. For inducing changes in light regime, mice were placed in a light cycler (Park Bioservices) for a minimum of 2 weeks to completely establish a 12-h inverted light cycle. Under these conditions, ZT5 corresponded to 8:30 pm and ZT13 to 12:30 pm. In additional experiments, mice subjected to MI at ZT13 were randomized in two groups to receive monoclonal neutrophil-depleting antibody (clone 1A8; 50 µg; BioXcell) or isotype by intraperitoneal (i.p.) injection 30 min after LAD occlusion and then every 24 h for up to 7 days. In other experiments, mice received i.p. injection of CXCR2 antagonist SB 225002 (Tocris, 1 mg/kg) or vehicle 5 min before LAD occlusion. Further experiments were performed in which mice were subjected to transient 45-min

ischemia followed by 24 h of reperfusion. Five min before reopening the occluded LAD, mice received an i.p. injection of SB 225002 (1 mg/kg) or vehicle. All animal experimental procedures were performed in strict accordance with the Guide for the Care and Use of Laboratory Animals published by the US National Institutes of Health (NIH publication No. 85-23, revised 1996) and were approved by the local ethnical committee (District Government of Upper Bavaria).

### Conditional inactivation of CXCR2 in mice and generation of bone marrow chimeras

We induced homologous recombination in embryonic stem (ES) cells using a construct containing 4.9 kb of gDNA, including exon 1 and part of intron 1–2, an IRES-LacZ cassette and Neo Cassette were inserted in between Frt sites, a loxP-site was introduced in 5′ direction to the Neo cassette, and 2.9 kb of gDNA, including CXCR2 exons 2 and 3, was flanked by loxP sites (floxed). The construct was finished by introducing a 3.5-kb fragment of gDNA serving as 3′ recombination arm (Appendix Fig S3). Correctly targeted ES cell clones were injected into blastocysts to produce a chimeric mouse that transmitted the modified allele through the germ line. A male heterozygous for the targeted allele was bred with a female expressing ubiquitous Flippase (Flp) transgene to ultimately produce animals that had deleted the IRES-LacZ and Neo cassettes, preserving the loxP sites flanking exons 2 and 3. Those homozygous animals were bred with mice expressing a MRP8-Cre transgene (Passegue *et al*, 2004) to ultimately produce animals that had deleted CXCR2 coding exons into neutrophils.

C57BL/6J wild-type mice (Janvier) underwent lethally whole body irradiation (2 × 5 Gy). Donor bone marrow cells were obtained from Mrp8-CreCXCR2flox or CXCR2flox mice (litermates). After flushing bones (femur, tibia, humerus), cells were washed, filtered, and intravenously injected (4 × 10$^6$ cells/mouse) in sterile saline into recipient mice 1 day after irradiation. Post-transplantation, recipient mice were reconstituted for another 6 weeks before undergoing LAD occlusion surgical procedure.

### Echocardiography

Transthoracic echocardiography was performed on mildly anesthetized spontaneously breathing mice (sedated by inhalation of 1% isoflurane, 1 l/min oxygen), using a Vevo® 2100 High Resolution Imaging system equipped with a 40-MHz transducer (VisualSonics, Toronto, Canada). The mice were placed on a heated ECG platform. Left parasternal long-axis view and left mid-papillary, apical and basal short-axis views were acquired. End-diastolic volume, end-systolic volume, and ejection fraction were evaluated on the left parasternal long-axis and parasternal short-axis view in a blinded manner.

### Infarct size and cause of death

Hearts were perfused and harvested 24 h after LAD ligation and sectioned into four equal transverse slices. The slices were incubated in 2% triphenyltetrazolium chloride (TTC) solution (Sigma-Aldrich) at 37°C for 15 min and fixed overnight in 4% formol at 4°C. For quantification of the area at risk, Evan's blue was injected into the left ventricle to distinguish between perfused cardiac tissue stained blue and non-perfused area at risk 24 h after MI. The area at risk was calculated as the percentage relative to the left ventricle (Appendix Fig S4). Images were taken at 10× magnification, and quantification of viable (red) and infarct areas (white) was performed in a blinded manner with ImageJ software. Each mouse which died after surgery prior to organ harvest underwent thoracotomy to investigate whether there was blood inside the pericardium indicating cardiac rupture.

### Histology

Four-micrometer paraffin sections were stained with Masson's trichrome (Sigma-Aldrich). Fibrosis was quantified as the relative area of blue staining (collagen) compared to the left ventricle surface, as an average of 3–4 sections per heart at the level of the papillary muscle, using ImageJ software. The anterior wall thickness of the left ventricle was measured on Masson's trichome-stained sections as an average of 3–4 sections per heart. For Sirius Red staining of collagen, 3–4 sections per heart were incubated with 0.1% Sirius Red (Sigma-Aldrich). Sections were photographed with identical exposure settings under ordinary polychromatic or polarized light microscopy. Total collagen content was evaluated under polychromatic light. Interstitial collagen subtypes were evaluated using polarized light illumination; under this condition, thicker type I collagen fibers appeared orange or red, whereas thinner type III collagen fibers were yellow or green. Quantifications were performed with LAS software (Leica). For quantification of myofibroblasts, sections were stained with an antibody against smooth muscle actin (αSMA, clone 1A4 Sigma-Aldrich, dilution 1/300). Myofibroblast density was quantified using ImageJ software by examining 10 fields per section at 20× magnification, in a blinded fashion.

### Cytokine and chemokine analysis

Blood was harvested by cardiac puncture, and bone marrow supernatant was obtained by flushing femurs three times with 2 ml of saline. Troponin I levels were measured with a precoated enzyme-linked immunosorbent assay (ELISA, Biotrend Chemicals). G-CSF, CXCL5, and CXCL12 in plasma and bone marrow supernatant were quantified with ELISA DuoSets from R&D systems. All other pro-inflammatory markers were quantified with ProcartaPlex™ Multiplex Immunoassay (eBioscience).

### Blood counter

Freshly obtained EDTA blood harvested by cardiac puncture was used to analyze leukocyte counts using an animal blood counter (scil Vet ABC Hematology Analyzer).

### Flow cytometry of heart and bone marrow

Hearts were harvested, perfused with saline to remove peripheral cells, minced with fine scissors, and digested with collagenase I (450 U/ml), collagenase XI (125 U/ml), hyaluronidase type I-s (60 U/ml), and DNase (60 U/ml; Sigma-Aldrich and Worthington Biochemical Corporation) at 37°C for 1 h. Bone marrow cells were obtained by flushing femurs with 2 ml of saline and triturated

through 70-μm nylon mesh strainer. The resulting single cell suspensions were centrifuged, resuspended in PBS/BSA 1%, and incubated with following monoclonal antibodies for 30 min at 4°C at 1/1,000 dilution: anti-CD45.2 (clone 104, BD Biosciences), anti-CD11b (clone M1/70, BioLegend), anti-c-Kit (clone 2B8, BioLegend), anti-Sca-1 (clone E13-161.7, BioLegend), anti-CD16/32 (clone 93, BioLegend), anti-CD34 (clone MEC14.7, BioLegend), anti-Ly6G (clone 1A8, BioLegend), lineage cocktail (clone 17A2/RB6-8C5/RA3-6B2/Ter-119/M1/70, BioLegend), and isotype controls (BioLegend). Anti-CD182 (CXCR2, PerCP/Cy5.5 labeled, clone TG11/CXCR2, BioLegend) was used at 1/300 dilution. Viable cells were identified as unstained with dead cell marker Zombie Yellow™ (BioLegend) in a 1/100 dilution. Data were acquired on a FACS Canto II (BD Biosciences), and analysis was performed with FlowJo software (Ashland, USA). Neutrophils were identified as $CD45^+$, $CD11b^+$, and $Ly6G^+$; granulocyte–monocyte progenitor cells (GMPs) were identified as $lineage^-$, $Sca-1^-$, $c-kit^+$, $CD16/32^+$, and $CD34^+$ (Fig 2); and dead cardiomyocytes were identified as $Zombie^+$, $CD45^-$. Gating for $CXCR2^{hi}$ was performed as shown in Appendix Fig S5.

### Quantitative real-time PCR

Whole RNA from lysed hearts (TissueLyser LT, Qiagen) was extracted (RNeasy mini kit, Qiagen) and reverse-transcribed (PrimeScript™ RT reagent kit, Clontech). Real-time PCR was performed with the 7900HT Sequence Detection System (Applied Biosystems) using the KAPA PROBE FAST Universal qPCR kit (Peqlab) and predesigned primer and probe mix (TaqMan® Gene Expression Assays, Life Technologies). Messenger RNA expression of markers of interest was normalized to HPRT, and the fold induction was calculated by the comparative $C_t$ method.

### Statistical analysis

Sample size for *in vivo* experiments was calculated in order to provide a statistical power > 85% for an $\alpha < 0.05$ in detecting a population effect size > 0.8. Comparisons between two groups of normally distributed and not connected data were performed using the unpaired Student's *t*-test. Multiple group comparisons were performed by one-way analyses of variance analyses (ANOVA, for one independent variable) followed by Tukey's multiple comparison tests or two-way ANOVA (for two independent variables) followed by Bonferroni *post hoc* test. Mortality was analyzed by log-rank test. All results are expressed as mean ± SEM. $P < 0.05$ was considered significant.

**Expanded View** for this article is available online.

### Acknowledgements

This work was funded in part by the FoeFoLe (Foerderung von Forschung und Lehre) program of the Ludwig-Maximilians-University Munich (to S.S. and M.J.S.), the Deutsche Forschungsgemeinschaft (STE-1053/5-1 to S.S.; SO876/6-1 and as part of the SFB1123 TP A6 to O.S., M.D.; B5 to O.S. and TP A1 to C.W.; Emmy-Noether grant SCHE 1645/2-1 and SFB 914 project B09), the German Centre for Cardiovascular Research (DZHK MHA VD1.2 to C.W., O.S. and doctoral fellowship to M.J.S.), the European Research Council ERC (AdG 249929 to C.W. and ERC starting grant CIRCODE to C.S.), the NWO (VIDI project 91712303 to O.S.), and TransCard Ph.D. Fellowship (Helmholtz International

## The paper explained

### Problem

Mounting evidence suggests that the outcome after myocardial infarction (MI) is time-of-day dependent. This is supported by clinical studies reporting a correlation between infarct size assessed by cardiac enzymes and the time-of-day of ischemia onset. The identification of underlying mechanisms behind this association might help developing more specialized therapies to improve prognosis after MI.

### Results

We found that the heart represents an immunologically dynamic organ with circadian fluctuations of adhesion molecule and chemokine expression and recruited leukocytes. We observed that neutrophil production and retention in the bone marrow are time-of-day dependent and that circulating neutrophils at the beginning of the active phase have higher capacity to migrate into the myocardium due to upregulated CXCR2 expression. We conclude that MI onset during the peak phase of $CXCR2^{hi}$-expressing blood neutrophils and consequently cardiac neutrophil accumulation promotes an exaggerated inflammatory response. Reducing neutrophilic inflammation by blocking CXCR2 or partially depleting neutrophils prevents excessive cardiac damage after MI occurring during the active phase.

### Impact

Our findings provide a mechanistic explanation for the worse outcomes in patients with MI occurring during the sleep-to-wake transition period. Limiting exaggerated neutrophil-mediated inflammation in these patients might improve their clinical outcome after MI.

Research School "Translational Cardiovascular and Metabolic Medicine" to K.N.). We thank Donato Santovito for careful revision and discussion of the manuscript and Lusine Saroyan, Orsolya Kimbu Wade, Cornelia Seidl, Yvonne Jansen, and Patricia Lemnitzer for excellent technical assistance.

## Author contributions

MJS and MH performed surgical procedures and echocardiography recordings. MJS and MH performed histology, flow cytometry, plasma measurements, and gene expression analysis and analyzed data. KN performed flow cytometric measurements and analyzed data. KB generated CXCR2flox mice. MD generated bone marrow chimeras. OS, JD, CS, MD, CW, and SS designed experimental approaches. OS, CW, CS, and SS provided substantial funding. All authors discussed results and commented on the manuscript. MJS and SS wrote the manuscript.

## Conflict of interest

The authors declare that they have no conflict of interest.

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
