## [Review Process File · EMBO Molecular Medicine]

The time-of-day of myocardial infarction onset affects healing through oscillations in cardiac neutrophil recruitment

Maximilian J. Schloss, Michael Horckmans, Katrin Nitz, Johan Duchene, Maik Drechsler, Kiril Bidzhekov, Christoph Scheiermann, Christian Weber, Oliver Soehnlein, Sabine Steffens

Corresponding author: Sabine Steffens, Institute for Cardiovascular Prevention IPEK, LMU

Review timeline:

Submission date:	20 November 2015
Editorial Decision:	22 December 2016
Revision received:	22 March 2016
Editorial Decision:	18 April 2016
Revision received:	21 April 2016

Transaction Report:

Editor: Roberto Buccione

1st Editorial Decision

22 December 2016

Thank you for the submission of your manuscript to EMBO Molecular Medicine.

In this case we also experienced unusual difficulties in securing three willing and appropriate reviewers, also due to the overlap with the vacation period. As a further delay cannot be justified I have decided to proceed based on the two available consistent evaluations.

Both Reviewers are generally positive on your manuscript although they raise some issues that require your action. I will not dwell into much detail as their comments are detailed. I would like, however, to highlight a few main points.

Reviewer 1, as you will see, is unconvinced that you can exclude other Zeitgeber time dependent factors, and therefore would like you to verify the effect of reversion of the mouse circadian rhythm for MI. S/he also inquires as the fate of neutrophils and macrophages overday.

Reviewer 2 also suggests a number of actions to improve the manuscript. These include disagreement on the I/R model used and also notes incomplete circadian analysis. S/he also notes that it would be of greater clinical relevance to evaluate the effects of the CXCR2 antagonist after occlusion to increase clinical relevance. This reviewer also lists a number of other very important issues that impinging on the overall clinical relevance and conclusiveness of the findings, both very important for our title.

In conclusion, while publication of the paper cannot be considered at this stage, we would be willing

to consider a substantially revised submission, with the understanding that the Reviewers' concerns must be addressed with additional experimental data where appropriate and that acceptance of the manuscript will entail a second round of review. Specifically, while we would not be necessarily asking you to re-perform all experiments in a different I/R model, all the other points are well taken and must be fully addressed.

Please note that it is EMBO Molecular Medicine policy to allow a single round of revision only and that, therefore, acceptance or rejection of the manuscript will depend on the completeness of your responses included in the next, final version of the manuscript.

As you might know, EMBO Molecular Medicine has a "scooping protection" policy, whereby similar findings that are published by others during review or revision are not a criterion for rejection. However, I do ask you to get in touch with us after three months if you have not completed your revision, to update us on the status. Please also contact us as soon as possible if similar work is published elsewhere.

EMBO Molecular Medicine now requires a complete author checklist (<http://embomolmed.embopress.org/authorguide#editorial3>) to be submitted with all revised manuscripts. Provision of the author checklist is mandatory at revision stage; The checklist is designed to enhance and standardize reporting of key information in research papers and to support reanalysis and repetition of experiments by the community. The list covers key information for figure panels and captions and focuses on statistics, the reporting of reagents, animal models and human subject-derived data, as well as guidance to optimise data accessibility.

I also suggest that you carefully adhere to our guidelines for publication in your next version, including presentation of statistical analyses and our new requirements for supplemental data (see also below) to speed up the pre-acceptance process in case of a favourable outcome.

I look forward to seeing a revised form of your manuscript as soon as possible.

***** Reviewer's comments *****

Referee #1 (Remarks):

This is very interesting and provocative study showing time of day-dependent oscillations of cardiac neutrophil recruitment modulating cardiac remodeling in an MI model. The study is of interest to the general scientific community.

Whereas the authors clearly showed ZT dependent outcome, it is unclear if other ZT dependent, model independent factors are contributing to the findings. The authors should reverse the murine circadian rhythm and perform the MI at ZT5, 21 o'clock (9pm, time of the day) and ZT13 at 12 pm (noon, time of the day).

the authors describe an oscillations of neutrophils in uninjured hearts. What goes in must come out. How does the neutrophil number decrease over the day?
Is the same true for macrophages?

Referee #2 (Comments on Novelty/Model System):

As suggested in the comments to the authors a I/R model and treatment before reperfusion would have more clinical impact.

Referee #2 (Remarks):

Schloss and colleagues studies circadian oscillations of neutrophil levels under baseline conditions

and upon recruitment to the heart which determines infarct size, healing and cardiac function after MI. Especially cardiac neutrophil recruitment during the active phase MI at ZT13 resulted in significantly higher cardiac neutrophil infiltration and could be inhibited by CXCR2 antagonism. They therefore concluded that limiting exaggerated neutrophilic inflammation at this time point significantly reduced the infarct size and thereby improved cardiac function.

Although the research questions are intriguing and interesting, detailed analysis is missing and more mechanistic insights and clinical relevance needed.

Major comments:

- Authors use a rodent model of permanent occlusion, which to this reviewer remains a major questions on why the areas at risk (AAR) are not affected completely when ligation was performed at different time-points and followed for 24h rhythm. A difference of 20% in LV infarcted area is extreme in a permanent model. (fig 3A) Moreover, ZT13 has a drop-out of 40% (survival graph) and the ZT5 only 5 %, which is extremely low for a permanent occlusion model. Hereby, the authors suggest that perfusion of the heart is directly influenced by this rhythm and affect cell death. By this big effect on infarct size, it is not clear why volumes do not differ more on long term ? The observed differences would have more impact if a more clinical relevant model of ischemia/reperfusion had been used.
 - Parameters and expression of several read-outs are lacking a full circadian analysis, e.g. 1B, moreover, a 48 h profile of genes like in panel 1D is needed to see a real rhythm since now it appeared expression is going up. Since rhythms are present both in the circulating cells as in organ receptor levels, which part determines the observed recruitments?
 - Have neutrophil levels been reported to fluctuate in human setting?
 - Why are volumes different between figure 3 and 4, maximal obtained values are not similar.
 - Neutrophils in the paper are properly defined and I really appreciate it they show the gating strategy for their flowcytometry analysis. It makes it clear to see how these were analyzed.
 - Reduction of CXCR2 impair extravasation of neutrophils, but it does not impair their homing and localization to the heart where they can still attach to the vessel wall and cause local inflammation there. Did they see any evidence of this? A specific knock-down or knock-out animal would be more appropriate to demonstrate mechanistic insights. When using these specific knock-down experiments, do the authors see still the effect on infarct size and remodelling?
 - Giving the antagonist 5 minutes before permanent occlusion makes the approach less relevant for a clinical translation. What will happen upon reperfusion injury in a clinical relevant approach.
- Minor
- CXCR2 is receptor for IL-8. Did the authors measure that cytokine in this study?
 - Rodents have an opposing circadian clock, being active during the night and at rest in day time. Current experiments have been performed at different time point but translation of these findings and rhythms were not discussed.
 - Figure 1A is it just neutrophils that show this circadian pattern, or does the whole white blood cell population oscillate like this?
 - Figure 1B - Please indicate the other time points that are needed for a circadian effect.
 - Figure 2E - Why the levels of cxcl2 go down after MI at ZT5? After injury, one would still expect increase in neutrophil chemotaxis independent of time point (figure 2A)...
 - Figure 5A - Cells in the square are likely CXCR2hi. How did they define the difference between low and neg?

1st Revision - authors' response

22 March 2016

Response to reviewer's comments

Referee #1 (Remarks):

This is very interesting and provocative study showing time of day-dependent oscillations of cardiac neutrophil recruitment modulating cardiac remodeling in an MI model. The study is of interest to the general scientific community.

We are pleased to read that the article is of interest to the reviewer, and would like to thank the reviewer for his/her careful consideration and helpful comments regarding our manuscript. On the basis of the remarks provided by the reviewer, we have conducted additional experiments and analyses, which further strengthen the conclusions. We have revised the manuscript accordingly, as described below.

Whereas the authors clearly showed ZT dependent outcome, it is unclear if other ZT dependent, model independent factors are contributing to the findings. The authors should reverse the murine circadian rhythm and perform the MI at ZT5, 21 o'clock (9pm, time of the day) and ZT13 at 12 pm (noon, time of the day).

As suggested by the reviewer, we performed additional experiments with mice housed under shifted light cycle conditions. Under these conditions, ZT13 MI was induced at noon, while ZT5 MI was induced in the evening (8pm). In this setting, we found comparable effects on cardiac neutrophil recruitment and cardiac damage. Mice subjected to MI at ZT13 had 2-fold higher neutrophil numbers, increased numbers of dead cardiomyocytes and plasma troponin levels than mice subjected to ZT5 MI (revised Figure 2D and Supplemental Figure 1). Thus, we can conclude that the reported effects on neutrophil recruitment and cardiac injury are indeed driven by a light-dependent circadian regulation.

The authors describe an oscillation of neutrophils in uninjured hearts. What goes in must come out. How does the neutrophil number decrease over the day?

Is the same true for macrophages?

Neutrophils have a short life span: their numbers decrease over the day because they become old, die and are eliminated. Neutrophils undergo daily cycles of release from the bone marrow into the circulation and subsequent recycling of aged neutrophils to the bone marrow for removal. The elimination of aged neutrophils in the bone marrow then provides a signal for regulating the homeostatic release of their own precursors (Casanova-Acebes et al, Cell 2013, 153: 1025-1035).

The release, migration into the tissue and subsequent trafficking back into the bone marrow is tightly regulated by chemokines and their cognate receptors. Bone marrow stromal cells produce CXCL12, which provides a retention signal for hematopoietic cells expressing high levels of CXCR4. Circadian reductions of CXCL12 in the bone marrow correlate with oscillations of neutrophils in the circulation (Mendez-Ferrer et al, Nature 2008, 452: 442-447). Young neutrophils released in the circulation express low levels of CXCR4. However, CXCR4 is again upregulated on aged neutrophils, which will lead to their return back into the bone marrow for elimination.

We found that the CXCR2 ligand/receptor axis also follows rhythmic cycles of expression levels in the myocardium and circulating neutrophils, respectively. CXCR2 expression levels are low on circulating neutrophils at ZT 5 at baseline (revised Figure 1E) and after ZT5 MI (Fig. 5A), when high numbers of aged CXCR4^{hi} CD62L^{lo} neutrophils are present in the circulation (Casanova-Acebes et al, Cell 2013, 153: 1025-1035). Conversely, circulating neutrophils at ZT 13 at baseline or after ZT13 MI, the time point with lowest numbers of aged neutrophils in the blood, express high levels of CXCR2 (revised Figure 1E and Fig. 5A). Our experiments with CXCR2 antagonist and neutrophil-specific CXCR2 knockout (revised Fig. 5F-H) confirm that the CXCR2 axis is mediating the circadian rhythm-dependent extend of neutrophil infiltration in the myocardium after MI.

As to the second question, the fate of macrophages is less clear. Under homeostatic conditions, tissue macrophages are thought to self-maintain via local proliferation without recruitment of blood monocytes. However, under inflammatory conditions, circulating monocytes are transiently recruited into the tissue where they differentiate into macrophages and thereby complement tissue-resident macrophages. Two monocyte subsets have been identified in the

mouse, classical (Ly6C^{hi}) and non-classical (Ly6C^{lo}, Geissmann et al, *Immunity* 2003, 19:71-82). Classical monocytes are produced in the bone marrow and originate from macrophage-dendritic cell precursors (MDPs). They are recruited to sites of inflammation, but in absence of inflammation return to the bone marrow and differentiate into non-classical monocytes (Varol et al. *J Exp Med* 2007, 204:171-180). Thus, under steady state, classical monocytes are short-lived precursors of non-classical monocytes, with a half-life of 19hr (Yona et al. *Immunity* 2013, 38:79-91). The non-classical (“patrolling”) monocytes crawl along the endothelium to survey its integrity (Auffray et al, *Science* 2007, 317: 666-670), and are thought to not give rise to tissue macrophages. Their lifespan is controlled by the availability of the Ly6C^{hi} monocyte precursors, ranging from a half-life of 2 days to 11 days in case of Ly6C^{hi} depletion (Yona et al. *Immunity* 2013, 38:79-91).

In agreement with recently published data (Scheiermann et al, *Immunity* 2012, 37:290-301; Nguyen et al, *Science* 2013,341: 1483-1488), we found that monocytes show daily oscillations in the blood. Given that in the heart, neutrophils outnumber the monocytes during the first few hours after MI, whereas monocyte numbers significantly increase only 3 days after MI, we focused on neutrophils in this study.

Referee #2 (Comments on Novelty/Model System):

As suggested in the comments to the authors a I/R model and treatment before reperfusion would have more clinical impact.

As suggested by the reviewer, we performed additional experiments in the I/R model in order to confirm key findings and to highlight a potential therapeutic relevance (see new Fig. 6 of revised manuscript).

Referee #2 (Remarks):

Schloss and colleagues studies circadian oscillations of neutrophil levels under baseline conditions and upon recruitment to the heart which determines infarct size, healing and cardiac function after MI. Especially cardiac neutrophil recruitment during the active phase MI at ZT13 resulted in significantly higher cardiac neutrophil infiltration and could be inhibited by CXCR2 antagonism. They therefore concluded that limiting exaggerated neutrophilic inflammation at this time point significantly reduced the infarct size and thereby improved cardiac function.

Although the research questions are intriguing and interesting, detailed analysis is missing and more mechanistic insights and clinical relevance needed.

We are pleased to read that the article is of interest to the reviewer, and would like to thank the reviewer for his/her careful consideration and helpful comments regarding our manuscript. On the basis of the remarks provided by the reviewer, we have conducted additional experiments and analyses, which further strengthen the conclusions and clarify the underlying mechanism. We have revised the manuscript accordingly, as described below.

Major comments:

- Authors use a rodent model of permanent occlusion, which to this reviewer remains a major questions on why the areas at risk (AAR) are not affected completely when ligation was performed at different time-points and followed for 24h rhythm. A difference of 20% in LV infarcted area is extreme in a permanent model. (fig 3A) Moreover, ZT13 has a drop-out of 40% (survival graph) and the ZT5 only 5 %, which is extremely low for a permanent occlusion model. Hereby, the authors suggest that perfusion of the heart is directly influenced by this rhythm and affect cell death. By this

big effect on infarct size, it is not clear why volumes do not differ more on long term ? The observed differences would have more impact if a more clinical relevant model of ischemia/reperfusion had been used.

The area at risk (AAR) is defined as the myocardial tissue within the vascular territory that is distal to the culprit lesion of the infarct-related artery. As the height of ligation of the left descending artery is independent from the time point of the operation, the area at risk should be the same. To verify our surgical procedure we quantified the AAR which confirmed a comparable AAR in both experimental groups ZT5 and ZT13 ($43,7 \pm 2.6\%$ vs. $45,2 \pm 2.9\%$, $n=4$, $P>0.05$; Supplemental Fig. 4). The proportion of the AAR that ultimately survives depends on various factors such as collateral flow, microvascular dysfunction, inflammatory responses and others. These factors are crucially dependent of an adequate inflammatory response creating a beneficial environment for the ischemic heart. Thus, an exaggerated inflammatory response counteracts myocardial salvage pathways within the AAR, thus promoting an infarct size expansion.

We agree that the difference of infarct size between both time points is impressive; however, a previous report has already reported a dramatic effect of the time-of-day of MI on infarct size (albeit in the I/R model). Following 1 day of reperfusion, hearts subjected to ischemia at ZT12 resulted in 3.5-fold increases in infarct size compared to hearts subjected to ischemia at ZT0 (Durgan et al, Circ Res 2010, 106: 546-550).

The mortality rate 14 day post MI is significantly higher in the ZT13 operated group (89,7% vs. 67,8%, $n=87$; $p<0.01$). Due to larger infarct sizes and impaired remodeling, mice operated at ZT13 suffer from a worsened heart function and face the risk of cardiac rupture. Our mortality rate is generally low, as we optimized the conditions for peri- and post-surgical care. We use an anesthesia which we antagonize at the end of the operation and from which the mice recover very well. After the operation specially prepared moisturized food is placed in the bottom of the cage which is accessible to the mice without any strains.

The infarct size determines the extent of ventricular dilatation and hereby the loss of heart function. We observed a difference between the end-diastolic volumes of ZT5 and ZT13 operated mice 14 days after MI of $25,1 \text{ mm}^3$ ($104,5 \pm 7,5 \text{ mm}^3$ vs. $129,6 \pm 9,1 \text{ mm}^3$, $n=6$; $p<0.01$). The end-diastolic volumes in the ZT13 operated mice are 20% larger than in the ZT5 group, which is a remarkable difference for a cardiovascular parameter. In addition, it should be taken into account that approximately 30% of the ZT13 operated mice do not survive until day 14 when the echocardiographic measurement would take place. These mice most likely suffer from extreme dilatation and their values are not detected due to their death prior to the measurement.

Finally, as suggested by the reviewer, we performed additional experiments in the I/R model in order to confirm our key findings and to highlight the CXCR2 axis as a potential therapeutic target (Fig. 6).

- Parameters and expression of several read-outs are lacking a full circadian analysis, e.g. 1B, moreover, a 48 h profile of genes like in panel 1D is needed to see a real rhythm since now it appeared expression is going up. Since rhythms are present both in the circulating cells as in organ receptor levels, which part determines the observed recruitments?

As requested by the reviewer, we now provide complete data for neutrophil counts from ZT1 to ZT21 in blood and heart and measured cardiac mRNA profiles over 48hr for full circadian analysis (see revised Fig. 1).

Moreover, the reviewer raises an important question: are oscillations in the tissue or the circulating leukocyte the critical determinant for enhanced neutrophil influx? Indeed, we found oscillations in both tissue expression levels as well as circulating neutrophil number and surface chemokine receptor expression levels. Remarkably, the peak of chemokine and adhesion molecule mRNA levels within hearts is somewhat delayed (ZT17) with regard to the peak of cardiac neutrophil counts (ZT13), whereas maximum CXCR2 expression on neutrophils is paralleled with their tissue accumulation at ZT13 (Fig. 1E). We therefore focused on the CXCR2 axis as crucial determinant for circadian rhythm-dependent cardiac neutrophil recruitment, and were able to confirm the neutrophil CXCR2-dependent mechanism by using both a selective CXCR2 antagonist as well as cell-specific knockout mice. We conclude that the neutrophil function rather than the tissue properties is the more critical determinant for circadian oscillations of neutrophil counts within the heart.

- Have neutrophil levels been reported to fluctuate in human setting?

Diurnal rhythms of leukocytes in human blood are well-described (Haus et al, *Chronobiol Int.* 1999, 16: 581-622). In both mice and humans, circadian oscillations of immune cells in the tissue peak during the active phase, possibly allowing the host to anticipate infections more efficiently (Scheiermann et al, *Nat Rev Immunol.* 2013, 13: 190-208). Oscillations in the blood are opposing to the tissue, peaking during the rest phase. Human blood neutrophils oscillate throughout the day with an amplitude of $0.31 \cdot 10^9/L$ and a high point around 8:30 pm (Sennels et al, *Scand J Clin Lab Invest.* 2011, 71: 532-541; see introduction p. 3 of the revised manuscript).

- Why are volumes different between figure 3 and 4, maximal obtained values are not similar.

In Fig. 3H, heart function and ventricular volumes of ZT5 and ZT13 operated mice was followed until day 14 after MI, whereas in Fig. 4G the measurement was performed only until day 7. Subsequently one has to compare the volumes at day 7: The end-diastolic volumes are around $75-100\text{mm}^3$ and end-systolic ventricular volumes around $55-80\text{mm}^3$, which is in the expected range.

- Neutrophils in the paper are properly defined and I really appreciate it they show the gating strategy for their flowcytometry analysis. It makes it clear to see how these were analyzed.

We appreciate that the reviewer is happy with our detailed information on the neutrophil gating strategy.

- Reduction of CXCR2 impair extravasation of neutrophils, but it does not impair their homing and localization to the heart where they can still attach to the vessel wall and cause local inflammation there. Did they see any evidence of this? A specific knock-down or knock-out animal would be more appropriate to demonstrate mechanistic insights. When using these specific knock-down experiments, do the authors see still the effect on infarct size and remodelling?

We agree that a cell-specific knockout gives more mechanistic insight than systemic blocking and performed additional experiments with a newly generated mouse strain that was not available from the beginning of our study. Indeed, the role of CXCR2 in neutrophil recruitment is not restricted to its expression on hematopoietic cells, but also endothelial CXCR2 expression is important for their migration into the tissue (Reutershan et al, *JCI* 2006, 116: 695-702). Thus, systemic administration of the CXCR2 antagonist may inhibit neutrophil recruitment by blocking CXCR2 both on neutrophils and endothelial cells in cardiac microvessels.

We did not find evidence for selective blockade of neutrophil extravasation at ZT13 and consequently enhanced local accumulation of neutrophils. The flow cytometric analysis of perfused, digested hearts includes both adherent and extravasating cells. Thus, the significant reduction of neutrophils in the heart (Fig. 5D) is more likely to reflect a reduced recruitment to the heart, not just inhibition of extravasation. From previous experiments based on systemic injection of a fluorochrome-labeled CD45 antibody shortly prior to organ harvest to label adherent, but not yet transmigrated cells, we know that these generally represent a minor number of leukocytes in the heart. Most of the leukocytes recruited to the heart few hours after MI have transmigrated.

Based on the bone marrow data shown in Fig. 5E, we believe that the CXCR2 antagonist-mediated reduction of cardiac neutrophil infiltration at ZT13 might be explained, at least in part, by reduced bone marrow mobilization. In support of this hypothesis, it has been previously shown that CXCR2 is required for bone marrow mobilization (Eash et al, JCI 2010).

To better clarify the mechanistic implication of CXCR2 in circadian-rhythm-dependent cardiac neutrophil recruitment, we performed additional experiments with mice carrying neutrophil-specific CXCR2 knockout (Mrp8Cre-CXCR2flox). Due to the restricted time frame for manuscript revision and very limited amount of available donor mice from our newly generated mouse line, we performed bone marrow chimeras. Thereby, we obtained 15 mice reconstituted with Mrp8-CXCR2flox and 18 control mice receiving CXCR2 WT bone marrow (CXCR2flox). Knockdown of CXCR2 expression by neutrophils was validated by flow cytometry. We found that neutrophil-selective CXCR2 knockout ablated excessive cardiac neutrophil recruitment after ZT13 MI (as shown in Fig. 5H) and infarct size as determined by quantification of troponin I plasma levels (see below: white bars represent WT controls, red bars represent CXCR2 KO, 24h post-MI, n = 6 for both groups at ZT5, n = 5 for CXCR2 KO at ZT13, n = 7 for WT at ZT13).

- Giving the antagonist 5 minutes before permanent occlusion makes the approach less relevant for a clinical translation. What will happen upon reperfusion injury in a clinical relevant approach.

We thank the reviewer for this valuable suggestion and performed an additional experiment by injecting the CXCR2 antagonist 5 min before reopening the occluded LAD. Using this

clinically more relevant approach, we were indeed able to show that CXCR2 blockade inhibits excessive neutrophil infiltration at ZT13 (see new Fig. 6).

Minor

- *CXCR2 is receptor for IL-8. Did the authors measure that cytokine in this study?*

As the human CXCR2 ligand IL-8 is not expressed in mice, we focused on functional murine homologues CXCL1, CXCL2 and CXCL5 (see Fig. 1 for baseline cardiac mRNA levels and Fig. 2 for plasma levels before and after MI).

- *Rodents have an opposing circadian clock, being active during the night and at rest in day time.*

Current experiments have been performed at different time point but translation of these findings and rhythms were not discussed.

We agree with the reviewer that this is a critical aspect and discussed this issue in the revised manuscript (introduction p. 3).

- *Figure 1A is it just neutrophils that show this circadian pattern, or does the whole white blood cell population oscillate like this?*

We have observed similar circadian oscillations in the blood at baseline with a peak at ZT5 and a trough at ZT13 for monocytes ($0.16 \times 10^9 \pm 0.02$ vs. $0.04 \times 10^9 \pm 0.01$; $n=6$; $p<0.01$) and lymphocytes ($4.6 \times 10^9 \pm 0.39$ vs. $1.4 \times 10^9 \pm 0.21$; $n=5-6$; $p<0.01$). However, neutrophils are the first innate immune responders recruited to the heart 24 hours after MI and are numerically the highest population in the blood and heart at this time point. As infarct size and chemokine secretion after 24 hours differed significantly between the ZT5 and ZT13 operated mice, we focused on neutrophils in this study.

- *Figure 1B - Please indicate the other time points that are needed for a circadian effect.*

Additional time points were added to revised Fig. 1B, confirming the trough for neutrophils in the heart at ZT5 and highest numbers at ZT13.

- *Figure 2E - Why the levels of cxcl2 go down after MI at ZT5? After injury, one would still expect increase in neutrophil chemotaxis independent of time point (figure 2A)...*

The CXCL2 plasma levels at baseline ZT5 and 24h after ZT5 MI are not significantly different ($p>0,05$, assessed by two-way ANOVA with Bonferroni post-hoc test). It is conceivable that increases in CXCL2 levels might occur rather locally in the heart than systemically in the plasma, which we are unable to detect at the protein level.

- *Figure 5A - Cells in the square are likely CXCR2hi. How did they define the difference between low and neg?*

To distinguish between CXCR2neg and CXCR2low neutrophils we used FMO controls. For clarification representative FACS plots including FMO controls are now shown in Supplemental Fig. 5 of the revised manuscript.

Thank you for the submission of your revised manuscript to EMBO Molecular Medicine. We have now received the enclosed reports from the referees that were asked to re-assess it. As you will see the reviewers are now globally supportive, Before I am able to process your manuscript further however, I must ask you to deal with the following final amendments:

- 1) While performing our pre-acceptance quality control and image screening routines, we noticed some occurrences of excessive background cleaning. I therefore ask you to please provide us with an explanation for this occurrence and original images.
- 2) As per our Author Guidelines, the description of all reported data that includes statistical testing must state the name of the statistical test used to generate error bars and P values, the number (n) of independent experiments underlying each data point (not replicate measures of one sample), and the actual P value for each test (not merely 'significant' or ' $P < 0.05$ ').
- 3) The supplementary information file should be renamed as Appendix and provided as a PDF file, Also, please note as per our author guidelines on the presentation of supplementary information (<http://embomolmed.embopress.org/authorguide#expandedview>), the figures should be re-labeled as Appendix Figure S1, Appendix Figure S2.... and the related call-outs in the manuscript adjusted accordingly
- 3) We are now encouraging the publication of source data, particularly for electrophoretic gels and blots, with the aim of making primary data more accessible and transparent to the reader. Would you be willing to provide a PDF file per figure that contains the original, uncropped and unprocessed scans of all or at least the key gels used in the manuscript? The PDF files should be labeled with the appropriate figure/panel number, and should have molecular weight markers; further annotation may be useful but is not essential. The PDF files will be published online with the article as supplementary "Source Data" files. If you have any questions regarding this just contact me.
- 4) Every published paper now includes a 'Synopsis' to further enhance discoverability. Synopses are displayed on the journal webpage and are freely accessible to all readers. They include a short standfirst as well as 2-5 one sentence bullet points that summarise the paper. Please provide the synopsis including the short list of bullet points that summarise the key NEW findings. The bullet points should be designed to be complementary to the abstract - i.e. not repeat the same text. We encourage inclusion of key acronyms and quantitative information. Please use the passive voice. Please attach this information in a separate file or send them by email, we will incorporate it accordingly. You are also welcome to suggest a striking image or visual abstract to illustrate your article. If you do please provide a jpeg file 550 px-wide x 400-px high.

Please submit your revised manuscript within two weeks, although I would like to hear form you on item 1 above as soon as possible

***** Reviewer's comments *****

Referee #1 (Remarks):

The authors responded adequately to my suggestions.
No further comments

Referee #2 (Remarks):

The authors did a clear job by adding more information and details and added several experiments to highlight their findings. No further comments.

Please find enclosed our revised manuscript entitled 'The time-of-day of ischemia onset affects myocardial infarction healing and heart function through oscillations in cardiac neutrophil recruitment' by Schloss et al. for publication in EMBO Molecular Medicine. We performed final amendments as requested and hope that the manuscript is now deemed suitable for publication.

Here is the list of final amendments performed as requested:

1. The submitted images were revised with Photoshop to clear background and exclude pericardial tissue (not included in the quantification) from the myocardial tissue (included in the ventricular fibrosis quantification) and pericardial tissue (not included in the quantification). Since in most cases pericardial tissue was not removed during LAD-ligation and the occlusion was done through the pericardial tissue, part of the pericardial tissue will remain attached to the epicardial wall. If quantification would be done including the pericardial tissue this would lead to falsely higher fibrotic area. To conform with the guidelines, the representative images shown in Fig 3F were replaced by original images of stained sections without background and pericardial tissue removal.
2. We updated the Figure legends in order to provide exact P values, n and statistical tests for all data including statistical testing. All n refer to individual mice (not replicate measures of one sample), as stated in the legend.
3. The supplementary information file and Figures, including the supplementary Figure links within the manuscript, were labeled according to the guidelines.
4. The publication of source data for gels and blots is not applicable. No blots are included.
5. A short synopsis as well as a graphical abstract were added as separate files.

I affirm that all authors agree with the submission and that material submitted for publication neither has been previously reported nor is under consideration for publication elsewhere.

I thank you very much for your time and consideration, hope for a positive response and look forward to hearing from you in due course

Corresponding Author Name: Sabine Steffens

Manuscript Number: EMM-2015-06083